# Altered Tryptophan Metabolism on the Kynurenine Pathway in Depressive Patients with Small Intestinal Bacterial Overgrowth

**DOI:** 10.3390/nu14153217

**Published:** 2022-08-06

**Authors:** Cezary Chojnacki, Paulina Konrad, Aleksandra Błońska, Marta Medrek-Socha, Karolina Przybylowska-Sygut, Jan Chojnacki, Tomasz Poplawski

**Affiliations:** 1Department of Clinical Nutrition and Gastroenterological Diagnostics, Medical University of Lodz, 90-647 Lodz, Poland; 2Department of Pharmaceutical Microbiology and Biochemistry, Medical University of Lodz, 90-136 Lodz, Poland; 3Department of Molecular Genetics, Faculty of Biology and Environmental Protection, University of Lodz, 90-236 Lodz, Poland

**Keywords:** l-tryptophan, kynurenic pathway, small intestinal bacterial overgrowth, mood disorders

## Abstract

The causes of depression are diverse and are still not fully understood. Recently, an increasing role is attributed to nutritional and inflammatory factors. The aim of this study was to evaluate selected metabolites of the tryptophan kynurenine pathway in depressive patients with small intestinal bacterial overgrowth (SIBO). The study involved 40 healthy people (controls) and 40 patients with predominant small intestinal bacterial overgrowth (SIBO-D). The lactulose hydrogen breath test (LHBT) was performed to diagnose SIBO. The severity of symptoms was assessed using the Gastrointestinal Symptom Rating Scale (GSRS–IBS) and the Hamilton Depression Rating Scale (HAM-D). The concentration of tryptophan (TRP), kynurenine (KYN), kynurenic acid (KYNA), and quinolinic acid (QA) in urine was determined using an LC–MS/MS method, before and after cyclic treatment with an antibiotic drug, rifaximin, for three months. The number of intraepithelial lymphocytes (IELs) in the duodenum and small intestinal mucosa, fecal calprotectin (FC) and serum level of *C*-reactive protein (CRP) were also determined. In patients with SIBO, a higher level of KYN and QA were found as compared to the control group. These two groups also differed in KYN/TRP (higher in SIBO) and KYNA/KYN ratios (lower in SIBO). A positive correlation was found between HAM-D and the number of IELs and the level of FC. Treatment with rifaximin improves the kynurenic pathway, as well as abdominal and mental complaints. Therefore, small intestinal bacterial overgrowth can be a cause of abdominal symptoms, but also mental disorders.

## 1. Introduction

Mood disorders, especially depression, are very common, suffering in people of all ages. The causes of depression are complex but still not fully understood. It is suggested that the appearance of this disease is primarily affected by genetic factors, psychological factors, and atypical brain structure and function [1]. Recently, an increasingly important role is attributed to nutritional and inflammatory factors [2,3]. Attention has been drawn to tryptophan (TRP).

TRP is the basic substrate for the production of many bioactive compounds along four pathways. Two of them, the serotonin (SER and kynurenine (KYN)) pathways, are very important for central nervous system (CNS) function. In the SER pathway, with the participation of serotonin hydroxylase (TPH) in peripheral tissue (TPH-1) and in the CNS (TPH-2), TRP is converted to 5-hydroxytryptophan, and then to SER and melatonin (MEL). These last two act as antidepressants and improve sleep quality [4]. Most TRP (over 90%) is metabolized on the KYN pathway, with participation of indoleamine 2,3-dioxygenase (IDO), which is present in many human organs, including the CNS [5,6]. TRP is oxidized by IDO or tryptophan 2,3-dioxygenase (TDO), and the product of this reaction (*N*-formylkynurenine) is rapidly hydrolyzed to KYN. In subsequent steps, KYN is metabolized in two ways. In the neuroprotective branches, KYN is transformed into kynurenic acid (KYNA) by kynurenine aminotransferase. In the neurotoxic branch other metabolites, such as 3-hydroxykynurenine (3-HK), 3-hydroxyantranilic acid (3-HAA), picolinic acid (PA), anthranilic acid (AA), xanthurenic acid (XA), and quinolinic acid (QA), are formed (Figure 1). The proportion between neuroprotective and neurotoxic factors is important for the proper brain functions [7,8,9].

The mechanism of adverse effect of KYN compounds on the CNS is not exactly known. The generally accepted hypothesis assumes that its toxic effects are related to the production of free radicals and the negative influence of energy metabolism in nerve cells [10,11]. They cause disturbances in neurotransmission, as well as degeneration and apoptosis of astrocytes and microglial cells, as a consequence of these changes [12].

The neurotoxic effects of TRP metabolites provide a promising hypothesis for the SER–KYN hypothesis of depression [13,14]. According to this, depression can be caused by a deficiency of SER or an excess of KYN metabolites. A decrease in the blood level of SER was observed in several clinical trials in depressed patients and TRP supplementation was found beneficial [15,16,17]. This does not bring into question the above hypothesis, because only one percent of TRP is metabolized in the brain [18]. The main pool of SER comes from the digestive system and does not cross the blood–brain barrier [19]. However, different results were also obtained in research on the role of the KYN pathway. Takada et al. [20] found that the 5-HIAA/TRP ratio and the KYN/TRP ratio were not different between healthy controls and depressive patients. Kim and Jeon [21] has indicated that TRP catabolites in the KYN pathway that are unbalanced by stressor or inflammation induce SER that causes anxiety. Quak et al. [22] found no significant relation between depressive symptoms and the KYN/TRP ratio. Pompili et al. [23] also observed that the plasma concentration of 5-HIAA and the KYN metabolites did not differ between depressed and healthy people, except KYN, whose levels were lower in patients with depression. Colle et al. [24] found that SER, KYN, KYNA, PA, and XA were lower in patients with major depression. Ogyan et al. [25] conducted meta-analysis on metabolite levels of the KYN pathway in depressive patients. The authors determined that KYN and KYNA levels are decreased in patients with depression, while QA levels are increased. However, most authors investigated depression using different validated questionnaires and the type of depression was not sufficiently documented. Furthermore, usually concomitant gastrointestinal diseases have not been evaluated. This is important because most TRP is catabolized in the digestive system from where its metabolites are partially transported to the brain. TRP is metabolized by various cells, including hepatocytes, neuroendocrine cells, macrophages, monocytes, lymphocytes, and intestinal bacteria [26,27]. The number of bacteria and inflammatory cells vary in different states of the disease, including small intestinal bacterial overgrowth. Quantitative changes in bacteria can affect the functions of the microbiota–gut–brain axis, but also mental state, but the exact mechanism behind these processes is unknown [28,29].

The purpose of the study was to evaluate selected metabolites of the TRP–KYN pathway in depressive patients with small intestinal bacterial overgrowth.

## 2. Materials and Methods

### 2.1. Patients

The study focused on 80 subjects, aged 24–60 years, recruited in 2017–2021, and two groups were distinguished, 40 individuals each. The first group (control, Group I) included 40 healthy people, without any problems and not using drugs, and with negative results of the hydrogen breath test. Group II consists of patients with small intestinal bacterial overgrowth (SIBO), who show chronic diarrhoea, abdominal pain, and bloating, and with mild or moderate depression. SIBO patients were characterized by loose or watery stools, occurring >25% of the time, at least for six months.

The lactulose hydrogen breath test was performed to diagnose SIBO using a Gastrolyzer (Bedfont, Ltd., Harrietsham, UK). An increase in breath hydrogen after consuming 10 g lactulose by at least 20 ppm within 90 min during the test was currently adopted as a criterion for SIBO occurrence.

The severity of abdominal symptoms was assessed using Gastrointestinal Symptom Rating Scale–Irritable Bowel Syndrome—GSRS–IBS [30]. This scale includes 13 gastrointestinal symptoms for the last seven days. The elements measure the severity of abdominal pain (1), pain relieved by bowel action (2), bloating (3), passing gas (4), constipation (5), diarrhoea (6), loose stool (7), hard stools (8), urgent need for bowel movement (9), incomplete bowel emptying (10), fullness shortly after meal (11), fullness long after eating (12), and visible distension (13). Items are scored between 1 and 7 points.

At the beginning and the end of the study, participants completed the Hamilton Depression Rating Scale (HAM-D) [31]. European standards have been adopted for HAM-D: 10–19 points—mild anxiety/depression, 19–29 points—moderate anxiety/depression, over 30 points—severe anxiety/depression.

To determine other diseases of the GI tract, all patients underwent endoscopic and histological examination of the gastric, duodenal, jejunum, and terminal ileum as well as colonic mucosa. The average number of intraepithelial lymphocytes (IELs) in the duodenum and small intestinal mucosa was evaluated with the Ultra Vision Quanto Detection System (Immunologic BV, Duiven, The Netherlands).

Exclusion criteria included: H-pylori-induced gastritis, lymphocytic and ulcerative colitis, Crohn’s disease, allergy and food intolerance, liver and kidney diseases, diabetes, and the use of antibiotics, probiotics, and psychotropic drugs in the month prior to enrollment in the study.

### 2.2. Laboratory Tests

The following routine laboratory tests were performed: blood cell count, *C*-reactive protein, glucose, bilirubin, urea, creatinine, lipid profile, thyroid stimulating hormone, free thyroxine, free triiodothyronine, alanine and asparagine aminotransferases, gamma-glutamyltranspeptidase, alkaline phosphatase, amylase, lipase, antibodies to tissue transglutaminase, and deaminated gliadin peptide. The serum concentration of *C*-reactive protein (CRP) was determined by a latex agglutination photometric assay in COBAS INTEGRA 800 (Roche Diagnostic, Basel, Switzerland). Fecal calprotectin (FC) was evaluated by sandwich ELISA test in Quantum Blue Reader (Buhlman Diagnostics, Amherst, NH, USA). Excretion of the neutrophil cytosolic protein, calprotectin, in the stool is a validated measure of intestinal mucosal inflammation and thus, disease activity. Moreover, FC help to distinguish the noninflammatory conditions from inflammatory diseases. Thus, an increase in the stool concentration of this protein indirectly indicates bacterial overgrowth and it is related to bacterial overgrowth-related mood changes. The number of intraepithelial lymphocytes (IELs) in the duodenum and small intestinal mucosa was assessed with the Ultra Vision Quanto Detection System (Immunologic BV, Duiven, The Netherlands) computer program.

Urine samples for TRP and its metabolites were collected in the morning on an empty stomach into a special container with a 0.1% hydrochloric acid solution as a stabilizer. We determined the concentration of TRP and its following metabolites: KYN, KYNA, and QA using liquid chromatography with tandem mass spectrometry (LC–MS/MS in accordance with the manufacturer instructions (Ganzimmun Diagnostics AG, Mainz, Germany; D-ML-13147-01-01, accepted by European Parlaments—No 765/2008). The levels of these metabolites were expressed in mg per gram of creatinine (mg/gCr). The ratios of the levels of 5-HIAA and TRP as well as KYN and TRP were also calculated. The 5-HIAA/TRP ratio was considered an exponent of the activity of the SER pathway and the KYN/TRP ratio reflected the activity of the KYN pathway in TRP metabolism.

### 2.3. Therapeutic Procedures

After the clinical, laboratory, and breath tests were completed, treatment with antibiotic was recommended. Patients with SIBO were recommended to take rifaximin, in a daily dose of 1200 mg for 14 days, then 1200 mg for 10 days in the following two months. It was also recommended to maintain the current diet and not take any other medications, probiotics, or dietary supplements. Follow-up medical examinations with the assessment of relief of symptoms were performed after 1–3 months and laboratory and breath tests were repeated in the fourth month, after the end of the third treatment cycle.

### 2.4. Ethical Issues

This study was conducted in accordance with the Declaration of Helsinki and the principle of good clinical practice. Written consent was obtained from each subject enrolled in the study and the study protocol was approved by the Bioethics Committee of Medical University (RNN/176/18/KE).

### 2.5. Data Analysis

The normality of the data distribution was checked using the Shapiro–Wilk W test. The U Mann–Whitney test was used to compare difference between two groups. The correlations between the quantitative variables were analyzed using the Spearman rank test. Differences within groups before and after treatment were analyzed using the Wilcoxon signed rank test. All statistical analyzes were performed with STATISTICA 13.3 software (TIBCO Software Inc., Palo Alto, CA, USA).

## 3. Results

The general characteristics of the subjects included in the study, the selected laboratory data, mental state, and daily tryptophan intake are presented in Table 1.

The biochemical data in both groups had a similar value, with the exception of *C*-reactive protein and fecal calprotectin, which were higher in SIBO patients. Compared to the control group, significant differences were found between the results of the LHBT test and between the HAM-D score (*p* < 0.001 Table 1). Daily TRP consumption in the control group ranged from 940 mg to 1629 mg (1368 ± 193), and in patients from 1009 mg to 1728 mg (1428 ± 203).

The urinary levels of KYN and QA in SIBO patients were higher than in the control group patients: 0.61 ± 0.14 vs. 0.41 ± 0.12 mg/gCr (*p* < 0.05) and 4.22 ± 0.93 vs. 3.15 ± 0.91 mg/gCr (*p* < 0.001), respectively, while the level of KYNA was lower—2.10 ± 0.62 vs. 2.44 ± 0.50 mg/gCr (*p* < 0.001, Figure 2).

A positive correlation was found between the HAM-D score and the concentration of fecal calprotectin (*p* = 0.0105), as well as between the HAM-D score and the number of intraepithelial lymphocytes in the duodenum and intestinal mucosa (*p* = 0.0175, Table 2).

In the patients, unfavorable proportions between neurotoxic metabolites (KYN, QA) and neuroprotective metabolites (KYNA) were found; differences were statistically significant (Figure 3).

After treatment, all metabolite levels in the urine had been reduced, that is, KYN from 0.61 ± 0.14 to 0.53 ± 0.13 mg/gCr (*p* < 0.05), KYNA from 2.10 ± 0.62 to 1.99 ± 0.51 mg/gCr (*p* > 0.05), and QA from 4.22 ± 0.93 to 3.71 ± 069 mg/gCr (*p* < 0.001, Figure 4).

Similarly, the tryptophan metabolite ratios also changed favorably after rifaximin treatment: KYN/TRP decreased from 0.048 ± 0.01 to 0.04 ± 0.01 (*p* < 0.05), KYN/QA increased from 0.53 ± 0.21 to 0.57 ± 0.17 (*p* < 0.01); while KYN/KYNA had changed slightly, this was from 3.71 ± 1.38 to 4.22 ± 0.91 (*p* = 0.099, Figure 5).

After rifaximin treatment, the severity of abdominal symptoms decreased from 37.5 ± 3.76 points to 20.6 ± 3.62 points (*p* < 0.001, Figure 6). Diarrhoea resolved in 33 (82.5%) patients, abdominal pain in 32 (80.0%), bloating in 27 (67.5%), and flatulence in 26 (65.0%). There was also a reduction in depression symptoms from 15.5 ± 3.41 points to 10.2 ± 2.97 points (*p* < 0.001, Figure 5). However, in eight (20.0%) the mental mood, especially anxiety, continued. Rifaximin was well tolerated, but six (15.0%) patients experienced increased bloating and flatulence on certain treatment days.

## 4. Discussion

KYN is the central metabolite of the KYN pathway because it is the starting point for the stages of metabolites in which compounds with different properties are formed. The KYN pathway exists mainly in the liver, where it is responsible for over 90% of overall TRP degradation. In normal conditions, the KYN pathway also exists extrahepatically, but its contribution to TRP degradation is approximately 5–10%, but it becomes quantitatively more significant under conditions of immune–inflammatory system activation [32]. The important property of KYN is the ease of penetration of the blood–brain barrier [33]. In the CNS, approximately 60% of KYN comes from blood, and the rest is synthesized locally from the TRP pool [34]. The percentage of the peripheral pool may increase in functional and inflammatory diseases of the gastrointestinal tract. These changes coexist with abdominal complaints and mood disorders. Clarke et al. [35] observed increases in the KYN levels and the KYN/TRP ratio in patients. The authors interpreted their results as the consequence in an increase in IDO, an enzyme responsible for TRP degradation in the KYN pathway. Heikemper et al. [36] found a lower KYN/TRP ratio in the total IBS and IBS-diarrhoea groups compared to healthy people. Kaszthelyi et al. [37] showed that IBS patients had a higher blood concentration of SER and KYNA than healthy subjects. Fitzgerald et al. [38] showed positive correlation between the level of KYN/TRP and the intensity of IBS syndrome, as well as depressive syndromes. However, Christmas et al. [39] suggested that diarrhoea-predominant IBS would have elevated TRP in plasma due to changes in its metabolism, mainly enhanced the SER pathway, and inhibited the KYN pathway. These researchers suggest that the measurement of TRP metabolites may be useful as components of a biomarker panel to aid gastroenterologists in the diagnosis of IBS and other gastrointestinal disorders.

The results of the above studies do not explain the cause of the changes in TRP metabolism in patients with IBS, and the intestinal microbiota was not assessed in these studies. On the other hand, many studies show that SIBO occurs in most patients with IBS. Furthermore, other authors indicated different disturbances in the KYN pathway in people with inflammatory bowel diseases, under the influence of an increase in pro-inflammatory cytokines and IDO activity [40,41]. Increasing KYNA/TRP is closely associated with inflammation of the colonic mucosa. Increased levels of TRP metabolites, especially QA, indicated a high activity of TRP degradation in patients with active IBD [42,43]. Inflammatory cells, mainly macrophages, are a source of pro-inflammatory cytokines, as well as the neurotoxic QA [44]. Gupta et al. [45] indicated that IDO-1 expression in Crohn’s disease is associated with a lower serum tryptophan and a higher KYN/TRP ratio. However, the KYN/TRP ratio was positively correlated with CRP and with disease activity.

It is recognized that an increase in the level of CRP and pro-inflammatory cytokines is considered to be a pathogenetic agent of mood disorders. In our patients, no correlation was found between the level of CRP and the intensity of depression symptoms. The examination of pro-inflammatory cytokines was not the aim of the study. On the other hand, fecal calprotectin and the number of intraepithelial lymphocytes were tested to exclude Crohn’s disease. A modest increase in calprotectin levels as well as an increased number of intraepithelial lymphocytes may indicate a low-grade intensity inflammatory process in the small intestine, possibly as SIBO affecting. IELs are a source of cytokines; predominately CD8 and CD4 are involved in the activation of innate and acquired immunity. Furthermore, all subpopulations of IELs can migrate beyond the intestinal wall to the circulatory system and other organs, including the CNS, and involved various clinical manifestations [46,47]. They are also believed to possess the ability to produce SER and the biosynthesis of KYN compounds [48]. Our study has shown a positive correlation between the number of IELs, as well as fecal calprotectin, and the intensity of depressive symptoms. However, a low correlation was found between the LHBT results and HAM-D. However, SIBO patients had higher levels of KYN and QA compared to healthy people. The QA level showed a strong positive correlation with the LHBT test. Furthermore, in this group, the KYN/TRP ratio was higher, while the KYNA/KYN and KYNA/QA ratios were lower. These unfavorable changes were reduced after rifaximin treatment, which confirms the involvement of intestinal bacteria in tryptophan metabolism. The ratio of neuroprotective factors to neurotoxic factors (KYNA/QA) could be used to estimate the imbalance of TRP metabolism, in order to assess the degree of CNS dysfunction [6]. In healthy people QA cross the blood–brain barrier poorly but can alter this barrier and increase its crossing to the brain [12,29]. It is also credited with the possibility to the damage of endothelium of blood vessels. The metabolized TRP by endothelial IDO to KYN has been shown to the morphological changes in blood vessels in the processes of chronic inflammation [49].

Our results support the influence of intestinal bacteria on the degradation of TRP. Several bacterial species may directly convert TRP, mainly into indole and scatols [50]. These metabolites are suggested to activate the immune system through binding to aryl hydrocarbon receptors (AHR), stimulate gastrointestinal motility and secretion of intestinal hormones, as well as exert toxic effects on the systemic circulation. Other strains of bacteria capable of metabolizing TRP in the SER and KP. In addition, IDO-1 that initiates the SER pathway is found in many cells of the intestinal wall, including neuroendocrine cells, monocytes, and plasmocytic cells. IDO-1 is also expressed of various cells in the digestive tract, including macrophages, plasmocytic cells, monocytes, lymphocytes, as well as epithelial cells [51,52]. Several inflammatory diseases of the gastrointestinal tract are associated with an elevated expression of IDO-1. The results of our study indicate an upregulated expression of IDO also in patients with SIBO, which is manifested by a high KYN/TRP ratio. Alverado et al. [53] indicated that a high expression of IDO-1 in epithelial cells promotes secretory cell differentiation and mucus production. TPH-1 exerts similar effects, and as a consequence of these changes, chronic diarrhoea occurs. The root cause is an excess of bacteria in the small intestine in our patients. The proof of this is the resolution of diarrhoea, as well as the level of KYN following rifaximin treatment.

Unfortunately, there was no complete improvement in depression symptoms, and in 20% of these patients the change in mood did not reach statistical significance. This is probably due to the complex pathogenesis of mental disorders. It cannot be ruled out that in some patients the mood disorders were endogenous, and that bacterial and food factors adversely affected the course of depression. Many researchers recognize that raised inflammatory markers, such as CRP and pro-inflammatory cytokines, have a significant association with an altered KYN pathway and the subsequent development of depressive symptoms and might indicate a genetic vulnerability to these disorders [54,55]. It can be assumed that the KYN pathway represents one of the main points of the interaction between genetic and environmental factors involved in the pathogenesis of depression. It is possible that the increased expression of genes that produce pro-inflammatory cytokines may determine the genetic predisposition to mood disorders through increased activity of the KYN pathway. The expression of IDO-1 and the activity of the KYN pathway may increase in a variety of infections and inflammations. However, the increase in fecal calprotectin concentration found in our patients indicates that the cause of somatic and mental disorders are changes in the intestinal microbiome, which is confirmed by the results of antimicrobial treatment.

Finally, the slight increase in *C*-reactive protein and fecal calprotectin may indicate the participation of bacterial and inflammatory factors in the pathogenesis of depression. More research is needed. Future research is underway that can investigate whether depression is related to gut microbiota by splitting participants with depression into groups with and without overgrowth.

## 5. Conclusions

Small intestinal bacterial overgrowth alters TRP metabolism on the KYN pathway, which can be a cause of abdominal disorders and mood disorders.

## Figures and Tables

**Figure 1 nutrients-14-03217-f001:**
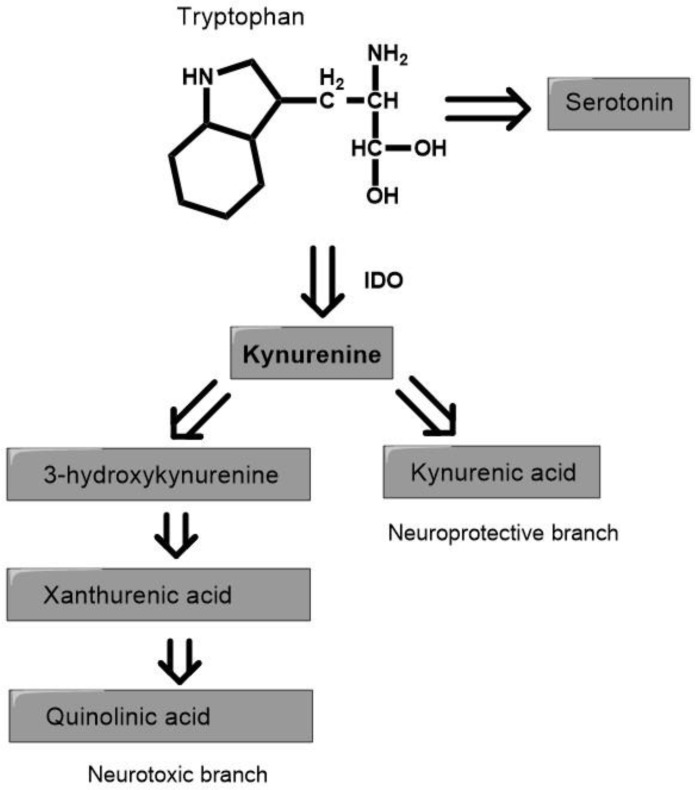
The simplified diagram of the pathways of tryptophan metabolism. IDO—indoleamine 2,3-dioxygenase.

**Figure 2 nutrients-14-03217-f002:**
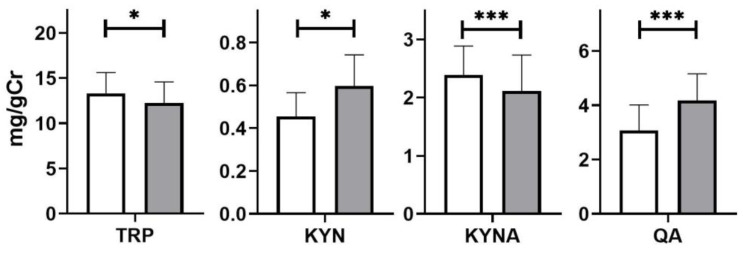
Urinary levels of tryptophan (TRP), kynurenine KYN, kynurenic acid (KYNA), and quinolinic acid (QA) in healthy subjects (white) and depressive patients with irritable bowel syndrome (blue). The differences between the groups were analyzed using the Mann-Whitney U test; n = 40 in both groups; * *p* < 0.05, *** *p* < 0.001.

**Figure 3 nutrients-14-03217-f003:**
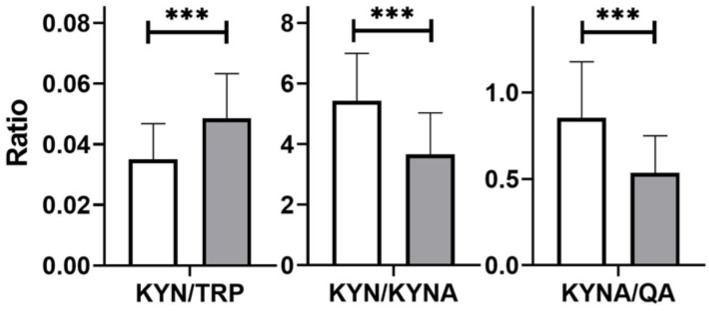
Ratios of tryptophan (TRP) and its metabolite levels (mg/gCr): kynurenine (KYN), kynurenic acid (KYNA), and quinolinic acid (QA) in healthy subjects (white) and in patients with small intestinal bacterial overgrowth (grey); differences between groups were analyzed using the Mann–Whitney U test; *n* = 40, *** *p* < 0.001.

**Figure 4 nutrients-14-03217-f004:**
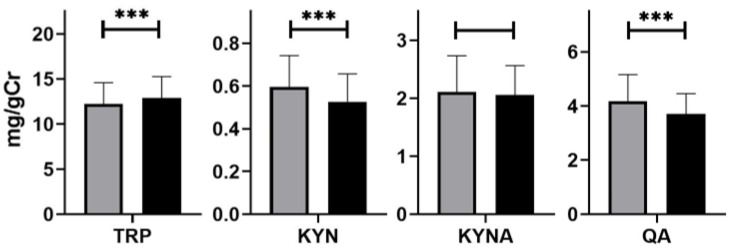
Urinary excretion of tryptophan (TRP) and its metabolites: kynurenine (KYN), kynurenic acid (KYNA), and quinolinic acid (QA) before (grey) and after (black) treatment with rifaximin. Differences were examined by Wilcoxon signed rank test; *n* = 40 in both groups; *** *p* < 0.001.

**Figure 5 nutrients-14-03217-f005:**
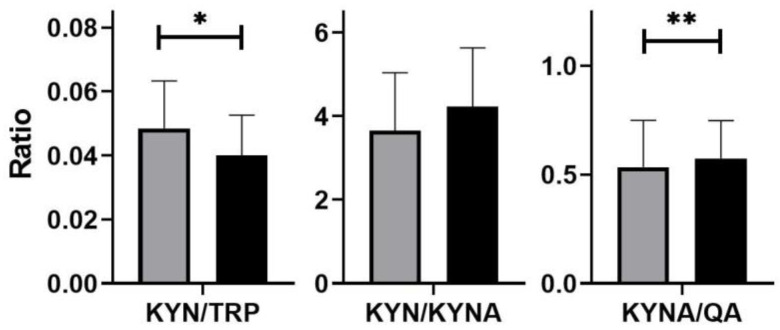
Ratios of tryptophan (TRP) and its metabolite levels (mg/gCr): kynurenine (KYN), kynurenic acid (KYNA), quinolinic acid (QA) in patients with small intestinal bacterial overgrowth before (grey) and after (black) rifaximin treatment. Differences were examined by Wilcoxon signed rank test; *n* = 40; * *p* < 0.05, ** *p* < 0.01.

**Figure 6 nutrients-14-03217-f006:**
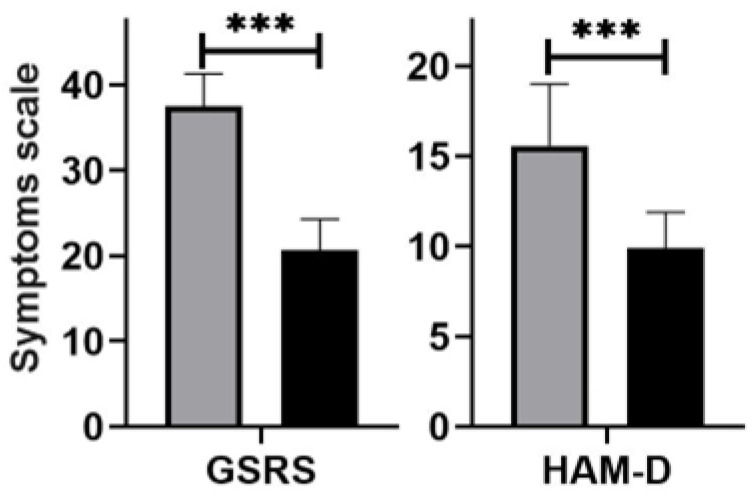
The core of the gastrointestinal symptoms rating scale (GSRS) and Hamilton Depression Rating Scale (HAM-D) score in depressive patients with small intestinal bacterial overgrowth before (grey) and after (black) rifaximin treatment. Differences were examined using the Wilcoxon signed rank test; *n* = 40; *** *p* < 0.001.

**Table 1 nutrients-14-03217-t001:** Characteristics of healthy subjects (Controls, *n* = 40) and patients with irritable bowel syndrome (SIBO, *n* = 40) included in the study, laboratory data, mental state, and tryptophan intake.

Feature	Controls	Patients
Age (years)	44.7 ± 7.3	45.2 ± 9.4
Gender-(M/F)	17/23	15/25
BMI (kg/m^2^)	23.8 ± 1.6	22.4 ± 2.1
CRP (mg/L)	1.59 ± 0.28	6.72 ± 2.91 *
FC (µg/g)	24.5 ± 6.4	46.2 ± 19.2 *
AST (U/L)	16.2 ± 3.1	17.1 ± 7.8
ALT (U/L)	17.4 ± 4.9	21.9 ± 7.6
Amylase (U/L)	35.7 ± 12.4	43.8 ± 19.2
Lipase (U/L)	72.6 ± 21.1	47.3 ± 15.4
Creatinine (mg/dL)	0.75 ± 0.23	0.86 ± 0.21
GFR (ml/min)	97.2 ± 10.3	88.9 ± 10.1
LHBT (ppm)	10.6 ± 4.19	69.2 ± 16.2 ***
HAM-D score	9.85 ± 2.21	26.3 ± 4.26 ***
TRP (mg daily)	1368 ± 193	1428 ± 203

Average ± SD (standard deviation), BMI—body mass index, CRP—*C*-reactive protein, FC—fecal calprotectin, AST—aspartate aminotransferase, ALT—alanine aminotransferase, GFR—glomerular filtration rate; LHBT—lactulose hydrogen breath test; HAM-D—Hamilton Depression Rating Scale; TRP—tryptophan intake; * *p* < 0.05; *** *p* < 0.001. Tables may have a footer.

**Table 2 nutrients-14-03217-t002:** Correlation between the Hamilton Depression Rating Scale (HAM-D) score and laboratory data; LHBT—hydrogen breath test, CRP—*C*-reactive protein, FC—fecal calprotectin, IEL—intraepithelial lymphocytes, KYN—kynurenine, KYNA—kynurenic acid, QA—quinolinic acid in 40 patients’ small intestinal bacterial overgrowth. The correlations were analyzed using the Spearman test. Significant *p*-values are highlighted in bold.

A Pairs of Variable	Rho-Spearman	*p*-Value
HAM-D and LHBT	**0.36024**	**0.05041**
HAM-D and CRP	0.27976	0.13616
HAM-D and FC	**0.33455**	**0.03486**
HAM-D and IEL	**0.37491**	**0.01752**
HAM-D and KYN	**0.35333**	**0.02545**
HAM-D and KYNA	0.33134	0.07480
HAM-D and QA	**0.50612**	**0.00430**

## Data Availability

The datasets generated during and/or analyzed during the current study are available from the corresponding authors on reasonable request.

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
