# Peer review of "Altered Tryptophan Metabolism on the Kynurenine Pathway in Depressive Patients with Small Intestinal Bacterial Overgrowth"

_nutrients, 2022, doi:10.3390/nu14153217_

Round 1

Reviewer 1 Report

Authors, this is a very interesting article potentially important clinical applications.  I had a few comments and questions. 

Line 55: wording, “In the neurotoxic branch another metabolites,..”.  Should it read, “In the neurotoxic branch, other metabolites,…”

Line 106:  would call it the “lactulose hydrogen breath test” for consistency

Lines 281-285.  Can you please describe in methods what an elevated fecal calprotectin is?  I don’t recall seeing a normal value.  Also, Crohn’s disease is normally diagnosed by more histological features than just IEL’s. Was a colonoscopy done to look at the terminal ileum?  Could the low grade intensity inflammation found in the small bowel be from SBBO? 

Line 292:  low correlation was found between LHBT results and HAM-D.  In your conclusion you state, "Small intestinal bacterial overgrowth alters tryptophan metabolism on the 340 kynurenine pathway, which can be a cause of abdominal disorders and mood disorders."

Have you looked at your data independent of those that had SBBO to see if those that were depressed had higher levels of QA, lower levels of KYNA compared to the controls?  Meaning, it's hard to know if the depression is coming from altered bacteria vs something else.  Would be good to compare patients with depression and split them up into groups (with and without depression). Not sure if I am making sense.  

Reviewer 2 Report

1.     The authors should specify in the abstract that rifaximin is an antibiotic drug. 

2.     Lines 30-33: the same sentence is repeated twice, please change it. 

3.     The introduction presented as a one long paragraph is confusing. Please divide it in shorter paragraphs organized by topic in order to facilitate the reading and understanding.

4.     Lines 94-96: the aim of the study should be rephrased as no inflammatory markers (e.g., cytokines) were measured and studied in correlation with the tryptophan kynurenine pathway. 

5.     The authors should carefully check the abbreviations along the text. When the use of abbreviation is needed make sure is specified at first use only (e.g., tryptophan).

6.     An additional figure showing the kynurenine pathway in physiological and proposed pathological condition should be added. 

7.     Line 117: The authors state “patients themselves assessed their mental health, and then everyone was assessed for mental health using the HAM-D”, how would a patient be able to assess objectively his/her mental health?

8.     LC-MS/MS method description and reference to the method should be provided.

9.     Table 2: I would suggest to highlight in bold significant p-values. 

10.  Based on the number of female and male subjects involved in the study, have the authors explored and observed any gender differences? If any, it might be worth to include the results in this study. 

11.  Organizing the discussion in shorter paragraphs would definitely help the reader. The current discussion organization as a one big paragraph makes it difficult to go through the author’s results overview and conclusions.

12.  Lines 284-285: low-grade intensity is repeated twice in the same sentence. 
